# Triple Generative Adversarial Nets

**Chongxuan Li, Kun Xu, Jun Zhu**[*]**, Bo Zhang**
Dept. of Comp. Sci. & Tech., TNList Lab, State Key Lab of Intell. Tech. & Sys.,
Center for Bio-Inspired Computing Research, Tsinghua University, Beijing, 100084, China
{licx14, xu-k16}@mails.tsinghua.edu.cn, {dcszj, dcszb}@mail.tsinghua.edu.cn

## Abstract

Generative Adversarial Nets (GANs) have shown promise in image generation and semi-supervised learning (SSL). However, existing GANs in SSL have two problems: (1) the generator and the discriminator (i.e. the classifier) may not be optimal at the same time; and (2) the generator cannot control the semantics of the generated samples. The problems essentially arise from the two-player formulation, where a single discriminator shares incompatible roles of identifying fake samples and predicting labels and it only estimates the data without considering the labels. To address the problems, we present triple generative adversarial net (Triple-GAN), which consists of three players—a generator, a discriminator and a classifier. The generator and the classifier characterize the conditional distributions between images and labels, and the discriminator solely focuses on identifying fake image-label pairs. We design compatible utilities to ensure that the distributions characterized by the classifier and the generator both converge to the data distribution. Our results on various datasets demonstrate that Triple-GAN as a unified model can simultaneously (1) achieve the state-of-the-art classification results among deep generative models, and (2) disentangle the classes and styles of the input and transfer smoothly in the data space via interpolation in the latent space class-conditionally.

## 1 Introduction

Deep generative models (DGMs) can capture the underlying distributions of the data and synthesize new samples. Recently, significant progress has been made on generating realistic images based on Generative Adversarial Nets (GANs) [7, 3, 22]. GAN is formulated as a two-player game, where the generator $G$ takes a random noise $z$ as input and produces a sample $G(z)$ in the data space while the discriminator $D$ identifies whether a certain sample comes from the true data distribution $p(x)$ or the generator. Both $G$ and $D$ are parameterized as deep neural networks and the training procedure is to solve a minimax problem:

$$\min_G \max_D U(D, G) = E_{x \sim p(x)}[\log(D(x))] + E_{z \sim p_z(z)}[\log(1 - D(G(z)))],$$

where $p_z(z)$ is a simple distribution (e.g., uniform or normal) and $U(\cdot)$ denotes the utilities. Given a generator and the defined distribution $p_g$, the optimal discriminator is $D(x) = p(x)/(p_g(x) + p(x))$ in the nonparametric setting, and the global equilibrium of this game is achieved if and only if $p_g(x) = p(x)$ [7], which is desired in terms of image generation.

GANs and DGMs in general have also proven effective in semi-supervised learning (SSL) [11], while retaining the generative capability. Under the same two-player game framework, Cat-GAN [26] generalizes GANs with a categorical discriminative network and an objective function that minimizes the conditional entropy of the predictions given the real data while maximizes the conditional entropy

---

[*]J. Zhu is the corresponding author.

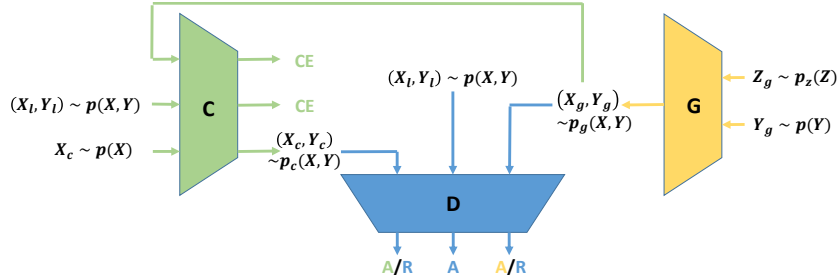

Figure 1: An illustration of Triple-GAN (best view in color). The utilities of $D$, $C$ and $G$ are colored in blue, green and yellow respectively, with "R" denoting rejection, "A" denoting acceptance and "CE" denoting the cross entropy loss for supervised learning. "A"s and "R"s are the adversarial losses and "CE"s are unbiased regularizations that ensure the consistency between $p_g$, $p_c$ and $p$, which are the distributions defined by the generator, classifier and true data generating process, respectively.

of the predictions given the generated samples. Odena [20] and Salimans et al. [25] augment the categorical discriminator with one more class, corresponding to the fake data generated by the generator. There are two main problems in existing GANs for SSL: (1) the generator and the discriminator (i.e. the classifier) may not be optimal at the same time [25]; and (2) the generator cannot control the semantics of the generated samples.

For the first problem, as an instance, Salimans et al. [25] propose two alternative training objectives that work well for either classification or image generation in SSL, but not both. The objective of *feature matching* works well in classification but fails to generate indistinguishable samples (See Sec.5.2 for examples), while the other objective of *minibatch discrimination* is good at realistic image generation but cannot predict labels accurately. The phenomena are not analyzed deeply in [25] and here we argue that they essentially arise from the two-player formulation, where a single discriminator has to play two incompatible roles—identifying fake samples and predicting labels. Specifically, assume that $G$ is optimal, i.e $p(x) = p_g(x)$, and consider a sample $x \sim p_g(x)$. On one hand, as a discriminator, the optimal $D$ should identify $x$ as a fake sample with non-zero probability (See [7] for the proof). On the other hand, as a classifier, the optimal $D$ should always predict the correct class of $x$ confidently since $x \sim p(x)$. It conflicts as $D$ has two incompatible convergence points, which indicates that $G$ and $D$ may not be optimal at the same time. Moreover, the issue remains even given imperfect $G$, as long as $p_g(x)$ and $p(x)$ overlaps as in most of the real cases. Given a sample form the overlapped area, the two roles of $D$ still compete by treating the sample differently, leading to a poor classifier[2]. Namely, the learning capacity of existing two-player models is restricted, which should be addressed to advance current SSL results.

For the second problem, disentangling meaningful physical factors like the object category from the latent representations with limited supervision is of general interest [30, 2]. However, to our best knowledge, none of the existing GANs can learn the disentangled representations in SSL, though some work [22, 5, 21] can learn such representations given full labels. Again, we believe that the problem is caused by their two-player formulation. Specifically, the discriminators in [26, 25] take a single data instead of a data-label pair as input and the label information is totally ignored when justifying whether a sample is real or fake. Therefore, the generators will not receive any learning signal regarding the label information from the discriminators and hence such models cannot control the semantics of the generated samples, which is not satisfactory.

To address these problems, we present Triple-GAN, a flexible game-theoretical framework for both classification and class-conditional image generation in SSL, where we have a partially labeled dataset. We introduce two conditional networks–a classifier and a generator to generate pseudo labels given real data and pseudo data given real labels, respectively. To jointly justify the quality of the samples from the conditional networks, we define a single discriminator network which has the sole role of distinguishing whether a data-label pair is from the real labeled dataset or not. The resulting model is called Triple-GAN because not only are there three networks, but we consider three joint distributions, i.e. the true data-label distribution and the distributions defined by the conditional networks (See Figure 1 for the illustration of Triple-GAN). Directly motivated by the desirable equilibrium that both the classifier and the conditional generator are optimal, we carefully design

compatible utilities including adversarial losses and unbiased regularizations (See Sec. 3), which lead to an effective solution to the challenging SSL task, justified both in theory and practice.

In particular, theoretically, instead of competing as stated in the first problem, a good classifier will result in a good generator and vice versa in Triple-GAN (See Sec. 3.2 for the proof). Furthermore, the discriminator can access the label information of the unlabeled data from the classifier and then force the generator to generate correct image-label pairs, which addresses the second problem. Empirically, we evaluate our model on the widely adopted MNIST [14], SVHN [19] and CIFAR10 [12] datasets. The results (See Sec. 5) demonstrate that Triple-GAN can simultaneously learn a good classifier and a conditional generator, which agrees with our motivation and theoretical results.

Overall, our main contributions are two folded: (1) we analyze the problems in existing SSL GANs [26, 25] and propose a novel game-theoretical Triple-GAN framework to address them with carefully designed compatible objectives; and (2) we show that on the three datasets with incomplete labels, Triple-GAN can advance the state-of-the-art classification results of DGMs substantially and, at the same time, disentangle classes and styles and perform class-conditional interpolation.

## 2   Related Work

Recently, various approaches have been developed to learn directed DGMs, including Variational Autoencoders (VAEs) [10, 24], Generative Moment Matching Networks (GMMNs) [16, 6] and Generative Adversarial Nets (GANs) [7]. These criteria are systematically compared in [28].

One primal goal of DGMs is to generate realistic samples, for which GANs have proven effective. Specifically, LAP-GAN [3] leverages a series of GANs to upscale the generated samples to high resolution images through the Laplacian pyramid framework [1]. DCGAN [22] adopts (fractionally) strided convolution layers and batch normalization [8] in GANs and generates realistic natural images.

Recent work has introduced inference networks in GANs. For instance, InfoGAN [2] learns explainable latent codes from unlabeled data by regularizing the original GANs via variational mutual information maximization. In ALI [5, 4], the inference network approximates the posterior distribution of latent variables given true data in unsupervised manner. Triple-GAN also has an inference network (classifier) as in ALI but there exist two important differences in the global equilibria and utilities between them: (1) Triple-GAN matches both the distributions defined by the generator and classifier to true data distribution while ALI only ensures that the distributions defined by the generator and inference network to be the same; (2) the discriminator will reject the samples from the classifier in Triple-GAN while the discriminator will accept the samples from the inference network in ALI, which leads to different update rules for the discriminator and inference network. These differences naturally arise because Triple-GAN is proposed to solve the existing problems in SSL GANs as stated in the introduction. Indeed, ALI [5] uses the same approach as [25] to deal with partially labeled data and hence it still suffers from the problems. In addition, Triple-GAN outperforms ALI significantly in the semi-supervised classification task (See comparison in Table. 1).

To handle partially labeled data, the conditional VAE [11] treats the missing labels as latent variables and infer them for unlabeled data. ADGM [17] introduces auxiliary variables to build a more expressive variational distribution and improve the predictive performance. The Ladder Network [23] employs lateral connections between a variation of denoising autoencoders and obtains excellent SSL results. Cat-GAN [26] generalizes GANs with a categorical discriminator and an objective function. Salimans et al. [25] propose empirical techniques to stabilize the training of GANs and improve the performance on SSL and image generation under incompatible learning criteria. Triple-GAN differs significantly from these methods, as stated in the introduction.

## 3   Method

We consider learning DGMs in the semi-supervised setting,[3] where we have a partially labeled dataset with $x$ denoting the input data and $y$ denoting the output label. The goal is to predict the labels $y$ for unlabeled data as well as to generate new samples $x$ conditioned on $y$. This is different from the unsupervised setting for pure generation, where the only goal is to sample data $x$ from a generator to fool a discriminator; thus a two-player game is sufficient to describe the process as in GANs.

In our setting, as the label information $y$ is incomplete (thus uncertain), our density model should characterize the uncertainty of both $x$ and $y$, therefore a joint distribution $p(x, y)$ of input-label pairs.

A straightforward application of the two-player GAN is infeasible because of the missing values on $y$. Unlike the previous work [26, 25], which is restricted to the two-player framework and can lead to incompatible objectives, we build our game-theoretic objective based on the insight that the joint distribution can be factorized in two ways, namely, $p(x, y) = p(x)p(y|x)$ and $p(x, y) = p(y)p(x|y)$, and that the conditional distributions $p(y|x)$ and $p(x|y)$ are of interest for classification and class-conditional generation, respectively. To jointly estimate these conditional distributions, which are characterized by a classifier network and a class-conditional generator network, we define a single discriminator network which has the sole role of distinguishing whether a sample is from the true data distribution or the models. Hence, we naturally extend GANs to Triple-GAN, a three-player game to characterize the process of classification and class-conditional generation in SSL, as detailed below.

## 3.1 A Game with Three Players

Triple-GAN consists of three components: (1) a classifier $C$ that (approximately) characterizes the conditional distribution $p_c(y|x) \approx p(y|x)$; (2) a class-conditional generator $G$ that (approximately) characterizes the conditional distribution in the other direction $p_g(x|y) \approx p(x|y)$; and (3) a discriminator $D$ that distinguishes whether a pair of data $(x, y)$ comes from the true distribution $p(x, y)$. All the components are parameterized as neural networks. Our desired equilibrium is that the joint distributions defined by the classifier and the generator both converge to the true data distribution. To this end, we design a game with compatible utilities for the three players as follows.

We make the mild assumption that the samples from both $p(x)$ and $p(y)$ can be easily obtained.[4] In the game, after a sample $x$ is drawn from $p(x)$, $C$ produces a pseudo label $y$ given $x$ following the conditional distribution $p_c(y|x)$. Hence, the pseudo input-label pair is a sample from the joint distribution $p_c(x, y) = p(x)p_c(y|x)$. Similarly, a pseudo input-label pair can be sampled from $G$ by first drawing $y \sim p(y)$ and then drawing $x|y \sim p_g(x|y)$; hence from the joint distribution $p_g(x, y) = p(y)p_g(x|y)$. For $p_g(x|y)$, we assume that $x$ is transformed by the latent style variables $z$ given the label $y$, namely, $x = G(y, z), z \sim p_z(z)$, where $p_z(z)$ is a simple distribution (e.g., uniform or standard normal). Then, the pseudo input-label pairs $(x, y)$ generated by both $C$ and $G$ are sent to the single discriminator $D$ for judgement. $D$ can also access the input-label pairs from the true data distribution as positive samples. We refer the utilities in the process as adversarial losses, which can be formulated as a minimax game:

$$\min_{C,G} \max_D U(C, G, D) = E_{(x,y)\sim p(x,y)}[\log D(x, y)] + \alpha E_{(x,y)\sim p_c(x,y)}[\log(1 - D(x, y))]$$

$$+ (1 - \alpha)E_{(x,y)\sim p_g(x,y)}[\log(1 - D(G(y, z), y))], \tag{1}$$

where $\alpha \in (0, 1)$ is a constant that controls the relative importance of generation and classification and we focus on the balance case by fixing it as $1/2$ throughout the paper.

The game defined in Eqn. (1) achieves its equilibrium if and only if $p(x, y) = (1 - \alpha)p_g(x, y) + \alpha p_c(x, y)$ (See details in Sec. 3.2). The equilibrium indicates that if one of $C$ and $G$ tends to the data distribution, the other will also go towards the data distribution, which addresses the competing problem. However, unfortunately, it cannot guarantee that $p(x, y) = p_g(x, y) = p_c(x, y)$ is the unique global optimum, which is not desirable. To address this problem, we introduce the standard supervised loss (i.e., cross-entropy loss) to $C$, $\mathcal{R}_\mathcal{L} = E_{(x,y)\sim p(x,y)}[-\log p_c(y|x)]$, which is equivalent to the KL-divergence between $p_c(x, y)$ and $p(x, y)$. Consequently, we define the game as:

$$\min_{C,G} \max_D \tilde{U}(C, G, D) = E_{(x,y)\sim p(x,y)}[\log D(x, y)] + \alpha E_{(x,y)\sim p_c(x,y)}[\log(1 - D(x, y))]$$

$$+ (1 - \alpha)E_{(x,y)\sim p_g(x,y)}[\log(1 - D(G(y, z), y))] + \mathcal{R}_\mathcal{L}. \tag{2}$$

It will be proven that the game with utilities $\tilde{U}$ has the unique global optimum for $C$ and $G$.

## 3.2 Theoretical Analysis and Pseudo Discriminative Loss

**Algorithm 1** Minibatch stochastic gradient descent training of Triple-GAN in SSL.

---

**for** number of training iterations **do**

    • Sample a batch of pairs $(x_g, y_g) \sim p_g(x, y)$ of size $m_g$, a batch of pairs $(x_c, y_c) \sim p_c(x, y)$ of size $m_c$ and a batch of labeled data $(x_d, y_d) \sim p(x, y)$ of size $m_d$.

    • Update $D$ by ascending along its stochastic gradient:

$$\nabla_{\theta_d} \left[ \frac{1}{m_d} (\sum_{(x_d, y_d)} \log D(x_d, y_d)) + \frac{\alpha}{m_c} \sum_{(x_c, y_c)} \log(1 - D(x_c, y_c)) + \frac{1 - \alpha}{m_g} \sum_{(x_g, y_g)} \log(1 - D(x_g, y_g)) \right].$$

    • Compute the unbiased estimators $\tilde{\mathcal{R}}_{\mathcal{L}}$ and $\tilde{\mathcal{R}}_{\mathcal{P}}$ of $\mathcal{R}_{\mathcal{L}}$ and $\mathcal{R}_{\mathcal{P}}$ respectively.

    • Update $C$ by descending along its stochastic gradient:

$$\nabla_{\theta_c} \left[ \frac{\alpha}{m_c} \sum_{(x_c, y_c)} p_c(y_c | x_c) \log(1 - D(x_c, y_c)) + \tilde{\mathcal{R}}_{\mathcal{L}} + \alpha_{\mathcal{P}} \tilde{\mathcal{R}}_{\mathcal{P}} \right].$$

    • Update $G$ by descending along its stochastic gradient:

$$\nabla_{\theta_g} \left[ \frac{1 - \alpha}{m_g} \sum_{(x_g, y_g)} \log(1 - D(x_g, y_g)) \right].$$

**end for**

---

We now provide a formal theoretical analysis of Triple-GAN under nonparametric assumptions and introduce the pseudo discriminative loss, which is an unbiased regularization motivated by the global equilibrium. For clarity of the main text, we defer the proof details to Appendix A.

First, we can show that the optimal $D$ balances between the true data distribution and the mixture distribution defined by $C$ and $G$, as summarized in Lemma 3.1.

**Lemma 3.1** *For any fixed $C$ and $G$, the optimal $D$ of the game defined by the utility function $U(C, G, D)$ is:*

$$D^*_{C,G}(x, y) = \frac{p(x, y)}{p(x, y) + p_\alpha(x, y)}, \tag{3}$$

*where $p_\alpha(x, y) := (1 - \alpha)p_g(x, y) + \alpha p_c(x, y)$ is a mixture distribution for $\alpha \in (0, 1)$.*

Given $D^*_{C,G}$, we can omit $D$ and reformulate the minimax game with value function $U$ as: $V(C, G) = \max_D U(C, G, D)$, whose optimal point is summarized as in Lemma 3.2.

**Lemma 3.2** *The global minimum of $V(C, G)$ is achieved if and only if $p(x, y) = p_\alpha(x, y)$.*

We can further show that $C$ and $G$ can at least capture the marginal distributions of data, especially for $p_g(x)$, even there may exist multiple global equilibria, as summarized in Corollary 3.2.1.

**Corollary 3.2.1** *Given $p(x, y) = p_\alpha(x, y)$, the marginal distributions are the same for $p$, $p_c$ and $p_g$, i.e. $p(x) = p_g(x) = p_c(x)$ and $p(y) = p_g(y) = p_c(y)$.*

Given the above result that $p(x, y) = p_\alpha(x, y)$, $C$ and $G$ do not compete as in the two-player based formulation and it is easy to verify that $p(x, y) = p_c(x, y) = p_g(x, y)$ is a global equilibrium point. However, it may not be unique and we should minimize an additional objective to ensure the uniqueness. In fact, this is true for the utility function $\tilde{U}(C, G, D)$ in problem (2), as stated below.

**Theorem 3.3** *The equilibrium of $\tilde{U}(C, G, D)$ is achieved if and only if $p(x, y) = p_g(x, y) = p_c(x, y)$.*

The conclusion essentially motivates our design of Triple-GAN, as we can ensure that both $C$ and $G$ will converge to the true data distribution if the model has been trained to achieve the optimum.

We can further show another nice property of $\tilde{U}$, which allows us to regularize our model for stable and better convergence in practice without bias, as summarized below.

**Corollary 3.3.1** *Adding any divergence (e.g. the KL divergence) between any two of the joint distributions or the conditional distributions or the marginal distributions, to $\tilde{U}$ as the additional regularization to be minimized, will not change the global equilibrium of $\tilde{U}$.*

Because label information is extremely insufficient in SSL, we propose *pseudo discriminative loss* $\mathcal{R}_{\mathcal{P}} = E_{p_g}[-\log p_c(y|x)]$, which optimizes $C$ on the samples generated by $G$ in the supervised manner. Intuitively, a good $G$ can provide meaningful labeled data beyond the training set as extra side information for $C$, which will boost the predictive performance (See Sec. 5.1 for the empirical evidence). Indeed, minimizing pseudo discriminative loss with respect to $C$ is equivalent to minimizing $D_{KL}(p_g(x,y)||p_c(x,y))$ (See Appendix A for proof) and hence the global equilibrium remains following Corollary 3.3.1. Also note that directly minimizing $D_{KL}(p_g(x,y)||p_c(x,y))$ is infeasible since its computation involves the unknown likelihood ratio $p_g(x,y)/p_c(x,y)$. The pseudo discriminative loss is weighted by a hyperparameter $\alpha_{\mathcal{P}}$. See Algorithm 1 for the whole training procedure, where $\theta_c$, $\theta_d$ and $\theta_g$ are trainable parameters in $C$, $D$ and $G$ respectively.

# 4 Practical Techniques

In this section we introduce several practical techniques used in the implementation of Triple-GAN, which may lead to a biased solution theoretically but work well for challenging SSL tasks empirically.

One crucial problem of SSL is the small size of the labeled data. In Triple-GAN, $D$ may memorize the empirical distribution of the labeled data, and reject other types of samples from the true data distribution. Consequently, $G$ may collapse to these modes. To this end, we generate pseudo labels through $C$ for some unlabeled data and use these pairs as positive samples of $D$. The cost is on introducing some bias to the target distribution of $D$, which is a mixture of $p_c$ and $p$ instead of the pure $p$. However, this is acceptable as $C$ converges quickly and $p_c$ and $p$ are close (See results in Sec.5).

Since properly leveraging the unlabeled data is key to success in SSL, it is necessary to regularize $C$ heuristically as in many existing methods [23, 26, 13, 15] to make more accurate predictions. We consider two alternative losses on the unlabeled data. The confidence loss [26] minimizes the conditional entropy of $p_c(y|x)$ and the cross entropy between $p(y)$ and $p_c(y)$, weighted by a hyperparameter $\alpha_{\mathcal{B}}$, as $\mathcal{R}_{\mathcal{U}} = H_{p_c}(y|x) + \alpha_{\mathcal{B}} E_p\big[-\log p_c(y)\big]$, which encourages $C$ to make predictions confidently and be balanced on the unlabeled data. The consistency loss [13] penalizes the network if it predicts the same unlabeled data inconsistently given different noise $\epsilon$, e.g., dropout masks, as $\mathcal{R}_{\mathcal{U}} = E_{x \sim p(x)}||p_c(y|x,\epsilon) - p_c(y|x,\epsilon')||^2$, where $||\cdot||^2$ is the square of the $l_2$-norm. We use the confidence loss by default except on the CIFAR10 dataset (See details in Sec. 5).

Another consideration is to compute the gradients of $E_{x \sim p(x), y \sim p_c(y|x)}[\log(1 - D(x,y))]$ with respect to the parameters $\theta_c$ in $C$, which involves summation over the discrete random variable $y$, i.e. the class label. On one hand, integrating out the class label is time consuming. On the other hand, directly sampling one label to approximate the expectation via the Monte Carlo method makes the feedback of the discriminator not differentiable with respect to $\theta_c$. As the REINFORCE algorithm [29] can deal with such cases with discrete variables, we use a variant of it for the end-to-end training of our classifier. The gradients in the original REINFORCE algorithm should be $E_{x \sim p(x)} E_{y \sim p_c(y|x)}[\nabla_{\theta_c} \log p_c(y|x) \log(1 - D(x,y))]$. In our experiment, we find the best strategy is to use most probable $y$ instead of sampling one to approximate the expectation over $y$. The bias is small as the prediction of $C$ is rather confident typically.

# 5 Experiments

We now present results on the widely adopted MNIST [14], SVHN [19], and CIFAR10 [12] datasets. MNIST consists of 50,000 training samples, 10,000 validation samples and 10,000 testing samples of handwritten digits of size $28 \times 28$. SVHN consists of 73,257 training samples and 26,032 testing samples and each is a colored image of size $32 \times 32$, containing a sequence of digits with various backgrounds. CIFAR10 consists of colored images distributed across 10 general classes—*airplane*, *automobile*, *bird*, *cat*, *deer*, *dog*, *frog*, *horse*, *ship* and *truck*. There are 50,000 training samples and 10,000 testing samples of size $32 \times 32$ in CIFAR10. We split 5,000 training data of SVHN and

Table 1: Error rates (%) on partially labeled MNIST, SHVN and CIFAR10 datasets, averaged by 10 runs. The results with $^{\dagger}$ are trained with more than 500,000 extra unlabeled data on SVHN.

| Algorithm | MNIST $n = 100$ | SVHN $n = 1000$ | CIFAR10 $n = 4000$ |
|---|---|---|---|
| *M1+M2* [11] | 3.33 ($\pm$0.14) | 36.02 ($\pm$0.10) | |
| *VAT* [18] | 2.33 | | 24.63 |
| *Ladder* [23] | 1.06 ($\pm$0.37) | | 20.40 ($\pm$0.47) |
| *Conv-Ladder* [23] | **0.89** ($\pm$0.50) | | |
| *ADGM* [17] | 0.96 ($\pm$0.02) | 22.86 $^{\dagger}$ | |
| *SDGM* [17] | 1.32 ($\pm$0.07) | 16.61($\pm$0.24)$^{\dagger}$ | |
| *MMCVA* [15] | 1.24 ($\pm$0.54) | **4.95** ($\pm$0.18) $^{\dagger}$ | |
| *CatGAN* [26] | 1.39 ($\pm$0.28) | | 19.58 ($\pm$0.58) |
| *Improved-GAN* [25] | 0.93 ($\pm$0.07) | 8.11 ($\pm$1.3) | 18.63 ($\pm$2.32) |
| *ALI* [5] | | 7.3 | 18.3 |
| *Triple-GAN* (**ours**) | **0.91** ($\pm$0.58) | **5.77**($\pm$0.17) | **16.99** ($\pm$0.36) |

Table 2: Error rates (%) on MNIST with different number of labels, averaged by 10 runs.

| Algorithm | $n = 20$ | $n = 50$ | $n = 200$ |
|---|---|---|---|
| *Improved-GAN* [25] | 16.77 ($\pm$4.52) | 2.21 ($\pm$1.36) | 0.90 ($\pm$0.04) |
| *Triple-GAN* (**ours**) | **4.81** ($\pm$4.95) | **1.56** ($\pm$0.72) | **0.67** ($\pm$0.16) |

CIFAR10 for validation if needed. On CIFAR10, we follow [13] to perform ZCA for the input of $C$ but still generate and estimate the raw images using $G$ and $D$.

We implement our method based on Theano [27] and here we briefly summarize our experimental settings.[5] Though we have an additional network, the generator and classifier of Triple-GAN have comparable architectures to those of the baselines [26, 25] (See details in Appendix F). The pseudo discriminative loss is not applied until the number of epochs reach a threshold that the generator could generate meaningful data. We only search the threshold in $\{200, 300\}$, $\alpha_{\mathcal{P}}$ in $\{0.1, 0.03\}$ and the global learning rate in $\{0.0003, 0.001\}$ based on the validation performance on each dataset. All of the other hyperparameters including relative weights and parameters in Adam [9] are fixed according to [25, 15] across all of the experiments. Further, in our experiments, we find that the training techniques for the original two-player GANs [3, 25] are sufficient to stabilize the optimization of Triple-GAN.

## 5.1 Classification

For fair comparison, all the results of the baselines are from the corresponding papers and we average Triple-GAN over 10 runs with different random initialization and splits of the training data and report the mean error rates with the standard deviations following [25].

Firstly, we compare our method with a large body of approaches in the widely used settings on MNIST, SVHN and CIFAR10 datasets given 100, 1,000 and 4,000 labels[6], respectively. Table 1 summarizes the quantitative results. On all of the three datasets, Triple-GAN achieves the state-of-the-art results consistently and it substantially outperforms the strongest competitors (e.g., Improved-GAN) on more challenging SVHN and CIFAR10 datasets, which demonstrate the benefit of compatible learning objectives proposed in Triple-GAN. Note that for a fair comparison with previous GANs, we do not leverage the extra unlabeled data on SVHN, while some baselines [17, 15] do.

Secondly, we evaluate our method with 20, 50 and 200 labeled samples on MNIST for a systematical comparison with our main baseline Improved-GAN [25], as shown in Table 2. Triple-GAN consistently outperforms Improved-GAN with a substantial margin, which again demonstrates the benefit of Triple-GAN. Besides, we can see that Triple-GAN achieves more significant improvement as the number of labeled data decreases, suggesting the effectiveness of the pseudo discriminative loss.

Finally, we investigate the reasons for the outstanding performance of Triple-GAN. We train a single $C$ without $G$ and $D$ on SVHN as the baseline and get more than $10\%$ error rate, which shows that $G$ is important for SSL even though $C$ can leverage unlabeled data directly. On CIFAR10, the baseline

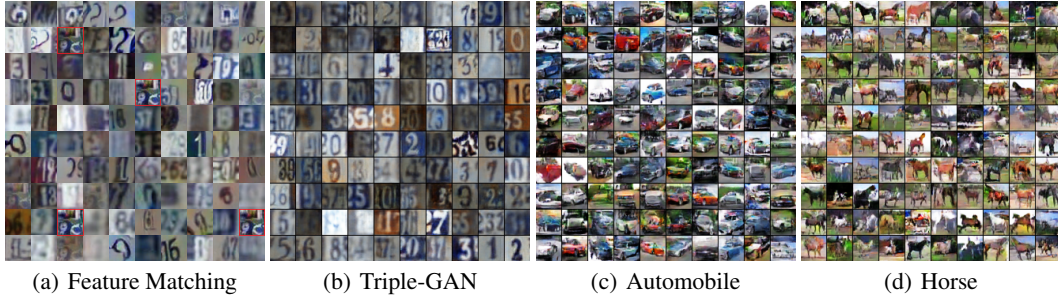

| (a) Feature Matching | (b) Triple-GAN | (c) Automobile | (d) Horse |

Figure 2: (a-b) Comparison between samples from Improved-GAN trained with feature matching and Triple-GAN on SVHN. (c-d) Samples of Triple-GAN in specific classes on CIFAR10.

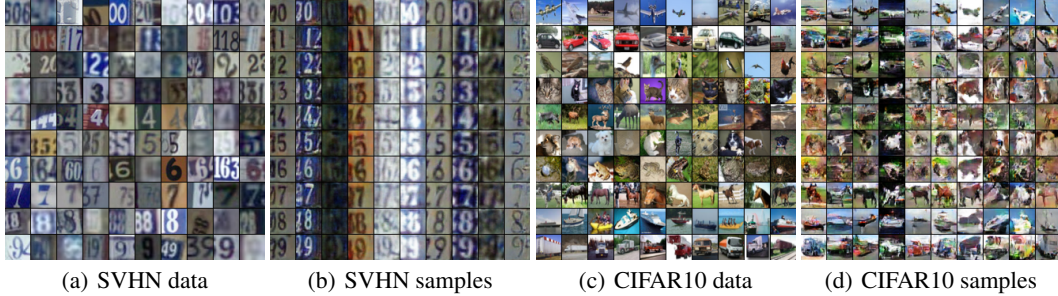

| (a) SVHN data | (b) SVHN samples | (c) CIFAR10 data | (d) CIFAR10 samples |

Figure 3: (a) and (c) are randomly selected labeled data. (b) and (d) are samples from Triple-GAN, where each row shares the same label and each column shares the same latent variables.

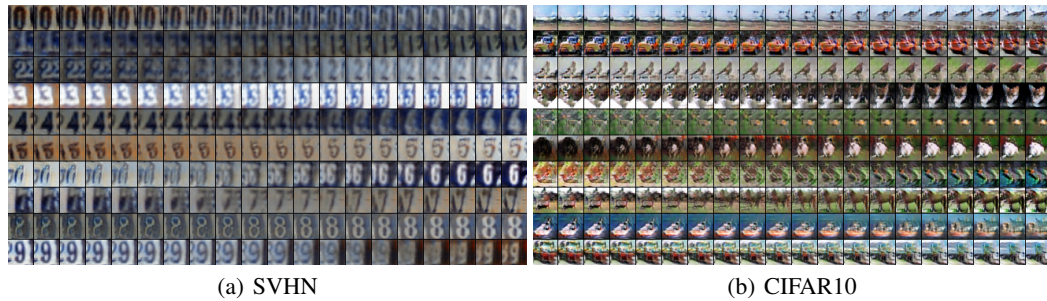

| (a) SVHN | (b) CIFAR10 |

Figure 4: Class-conditional latent space interpolation. We first sample two random vectors in the latent space and interpolate linearly from one to another. Then, we map these vectors to the data level given a fixed label for each class. Totally, 20 images are shown for each class. We select two endpoints with clear semantics on CIFAR10 for better illustration.

(a simple version of $\Pi$ model [13]) achieves 17.7% error rate. The smaller improvement is reasonable as CIFAR10 is more complex and hence $G$ is not as good as in SVHN. In addition, we evaluate Triple-GAN without the pseudo discriminative loss on SVHN and it achieves about 7.8% error rate, which shows the advantages of compatible objectives (better than the 8.11% error rate of Improved-GAN) and the importance of the pseudo discriminative loss (worse than the complete Triple-GAN by 2%). Furthermore, Triple-GAN has a comparable convergence speed with Improved-GAN [25], as shown in Appendix E.

## 5.2 Generation

We demonstrate that Triple-GAN can learn good $G$ and $C$ simultaneously by generating samples in various ways with the exact models used in Sec. 5.1. For fair comparison, the generative model and the number of labels are the same to the previous method [25].

In Fig. 2 (a-b), we first compare the quality of images generated by Triple-GAN on SVHN and the Improved-GAN with feature matching [25],[7] which works well for semi-supervised classification. We can see that Triple-GAN outperforms the baseline by generating fewer meaningless samples and

clearer digits. Further, the baseline generates the same strange sample four times, labeled with red rectangles in Fig. 2 . The comparison on MNIST and CIFAR10 is presented in Appendix B. We also evaluate the samples on CIFAR10 quantitatively via the inception score following [25]. The value of Triple-GAN is $5.08 \pm 0.09$ while that of the Improved-GAN trained without minibatch discrimination [25] is $3.87 \pm 0.03$, which agrees with the visual comparison. We then illustrate images generated from two specific classes on CIFAR10 in Fig. 2 (c-d) and see more in Appendix C. In most cases, Triple-GAN is able to generate meaningful images with correct semantics.

Further, we show the ability of Triple-GAN to disentangle classes and styles in Fig. 3. It can be seen that Triple-GAN can generate realistic data in a specific class and the latent factors encode meaningful physical factors like: scale, intensity, orientation, color and so on. Some GANs [22, 5, 21] can generate data class-conditionally given full labels, while Triple-GAN can do similar thing given much less label information.

Finally, we demonstrate the generalization capability of our Triple-GAN on class-conditional latent space interpolation as in Fig. 4. Triple-GAN can transit smoothly from one sample to another with totally different visual factors without losing label semantics, which proves that Triple-GANs can learn meaningful latent spaces class-conditionally instead of overfitting to the training data, especially labeled data. See these results on MNIST in Appendix D.

Overall, these results confirm that Triple-GAN avoid the competition between $C$ and $G$ and can lead to a situation where both the generation and classification are good in semi-supervised learning.

## 6 Conclusions

We present triple generative adversarial networks (Triple-GAN), a unified game-theoretical framework with three players—a generator, a discriminator and a classifier, to do semi-supervised learning with compatible utilities. With such utilities, Triple-GAN addresses two main problems of existing methods [26, 25]. Specifically, Triple-GAN ensures that both the classifier and the generator can achieve their own optima respectively in the perspective of game theory and enable the generator to sample data in a specific class. Our empirical results on MNIST, SVHN and CIFAR10 datasets demonstrate that as a unified model, Triple-GAN can simultaneously achieve the state-of-the-art classification results among deep generative models and disentangle styles and classes and transfer smoothly on the data level via interpolation in the latent space.

**Acknowledgments**

The work is supported by the National NSF of China (Nos. 61620106010, 61621136008, 61332007), the MIIT Grant of Int. Man. Comp. Stan (No. 2016ZXFB00001), the Youth Top-notch Talent Support Program, Tsinghua Tiangong Institute for Intelligent Computing, the NVIDIA NVAIL Program and a Project from Siemens.

## Footnotes

[2]The results of minibatch discrimination approach in [25] well support our analysis.

[3]Supervised learning is an extreme case, where the training set is fully labeled.

[4]In semi-supervised learning, $p(x)$ is the empirical distribution of inputs and $p(y)$ is assumed same to the distribution of labels on labeled data, which is uniform in our experiment.

[5]Our source code is available at https://github.com/zhenxuan00/triple-gan

[6]We use these amounts of labels as default settings throughout the paper if not specified.

[7]Though the Improved-GAN trained with minibatch discrimination [25] can generate good samples, it fails to predict labels accurately.

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
