[Supplementary Material]

# A    Detailed Theoretical Analysis

**Lemma 3.1.** *For any fixed $C$ and $G$, the optimal discriminator $D$ of the game defined by the utility function $U(C, G, D)$ is*

$$D^*_{C,G}(x, y) = \frac{p(x, y)}{p(x, y) + p_\alpha(x, y)},  \tag{1}$$

*where $p_\alpha(x, y) := (1 - \alpha)p_g(x, y) + \alpha p_c(x, y)$ is a mixture distribution for $\alpha \in (0, 1)$.*

*Proof.* Given the classifier and generator, the utility function can be rewritten as

$$U(C, G, D) = \iint p(x, y) \log D(x, y) dy dx + (1 - \alpha) \iint p(y) p_z(z) \log(1 - D(G(z, y), y)) dy dz$$

$$+ \alpha \iint p(x) p_c(y|x) \log(1 - D(x, y)) dy dx$$

$$= \iint p(x, y) \log D(x, y) dy dx + \iint p_\alpha(x, y) \log(1 - D(x, y)) dy dx = f(D(x, y)).$$

Note that the function $f(D(x, y))$ achieves the maximum at $\frac{p(x,y)}{p(x,y) + p_\alpha(x,y)}$.

$\square$

**Lemma 3.2.** *The global minimum of $V(C, G)$ is achieved if and only if $p(x, y) = p_\alpha(x, y)$.*

*Proof.* Given $D^*_{C,G}$, we can reformulate the minimax game with value function $U$ as:

$$V(C, G) = \iint p(x, y) \log \frac{p(x, y)}{p(x, y) + p_\alpha(x, y)} dy dx + \iint p_\alpha(x, y) \log \frac{p_\alpha(x, y)}{p(x, y) + p_\alpha(x, y)} dy dx.$$

Following the proof in GAN, the $V(C, G)$ can be rewritten as

$$V(C, G) = -\log 4 + 2 JSD(p(x, y) || p_\alpha(x, y)),  \tag{2}$$

where $JSD$ is the Jensen-Shannon divergence, which is always non-negative and the unique optimum is achieved if and only if $p(x, y) = p_\alpha(x, y) = (1 - \alpha)p_g(x, y) + \alpha p_c(x, y)$. $\square$

**Corollary 3.2.1.** *Given $p(x, y) = p_\alpha(x, y)$, the marginal distributions are the same for $p$, $p_c$ and $p_g$, i.e. $p(x) = p_g(x) = p_c(x)$ and $p(y) = p_g(y) = p_c(y)$.*

*Proof.* Remember that $p_g(x, y) = p(y) p_g(x|y)$ and $p_c(x, y) = p(x) p_c(y|x)$. Take integral with respect to $x$ on both sides of $p(x, y) = p_\alpha(x, y)$ to get

$$\int p(x, y) dx = (1 - \alpha) \int p_g(x, y) dx + \alpha \int p_c(x, y) dx,$$

which indicates that

$$p(y) = (1 - \alpha)p(y) + \alpha p_c(y), \text{ i.e. } p_c(y) = p(y) = p_g(y).$$

Similarly, it can be shown that $p_g(x) = p(x) = p_c(x)$ by taking integral with respect to $y$. $\square$

**Theorem 3.3.** *The equilibrium of $\tilde{U}(C, G, D)$ is achieved if and only if $p(x, y) = p_g(x, y) = p_c(x, y)$.*

*Proof.* According to the definition, $\tilde{U}(C, G, D) = U(C, G, D) + \mathcal{R}_\mathcal{L}$, where

$$\mathcal{R}_\mathcal{L} = E_p[-\log p_c(y|x)],$$

which can be rewritten as:

$$D_{KL}(p(x, y) || p_c(x, y)) + H_p(y|x).$$

Namely, minimizing $\mathcal{R}_\mathcal{L}$ is equivalent to minimizing $D_{KL}(p(x, y) || p_c(x, y))$, which is always non-negative and zero if and only if $p(x, y) = p_c(x, y)$. Besides, the previous lemmas can also be applied to $\tilde{U}(C, G, D)$, which indicates that $p(x, y) = p_\alpha(x, y)$ at the global equilibrium, concluding the proof. $\square$

|(a) Feature Matching|(b) Triple-GAN|(c) Feature Matching|(d) Triple-GAN|

Figure 1: (a) and (c): Samples generated from Improved-GAN trained with feature matching on MNIST and CIFAR10 datasets. Strange patterns repeat on CIFAR10. (b) and (d): Samples generated from Triple-GAN.

**Corollary 3.3.1.** *Adding any divergence (e.g. the KL divergence) between any two of the joint distributions or the conditional distributions or the marginal distributions, to $\tilde{U}$ as the additional regularization to be minimized, will not change the global equilibrium of $\tilde{U}$.*

*Proof.* This conclusion is straightforward derived by the global equilibrium point of $\tilde{U}$ and the definition of the divergence between distributions. $\qquad\square$

**Pseudo discriminative loss** We prove the equivalence of the pseudo discriminative loss in the main text and KL-divergence $D_{KL}(p_g(x,y)||p_c(x,y))$ as follows:

$$
\begin{aligned}
&D_{KL}(p_g(x,y)||p_c(x,y)) + H_{p_g}(y|x) - D_{KL}(p_g(x)||p(x)) \\
=&\iint p_g(x,y) \log \frac{p_g(x,y)}{p_c(x,y)} + p_g(x,y) \log \frac{1}{p_g(y|x)} dxdy - \int p_g(x) \log \frac{p_g(x)}{p(x)} dx \\
=&\iint p_g(x,y) \log \frac{p_g(x,y)}{p_c(x,y)p_g(y|x)} dxdy - \iint p_g(x,y) \log \frac{p_g(x)}{p(x)} dxdy \\
=&\iint p_g(x,y) \log \frac{p_g(x,y)p(x)}{p_c(x,y)p_g(y|x)p_g(x)} dxdy \\
=&E_{p_g}[-\log p_c(y|x)].
\end{aligned}
$$

Note that the last equality holds as $p_c(x) = p(x)$ and $H_{p_g}(y|x) - D_{KL}(p_g(x)||p(x))$ is a constant with respective to $\theta_c$. Therefore, if we only optimize $C$, these two losses are equivalent.

# B    Unconditional Generation

We compare the samples generated from Triple-GAN and Improved-GAN on the MNIST and CIFAR10 datasets as in Fig. 1, where Triple-GAN shares the same architecture of generator and number of labeled data with the baseline. It can be seen that Triple-GAN outperforms the GANs that are trained with the feature matching criterion on generating indistinguishable samples.

# C    Class-conditional Generation on CIFAR10

We show more class-conditional generation results on CIFAR10 in Fig. 2. Again, we can see that Triple-GAN can generate meaningful images in specific classes.

# D    Disentanglement and Interpolation on the MNIST dataset

We present the disentanglement of class and style and class-conditional interpolation on the MNIST dataset as in Fig. 3. We have the same conclusion as in main text that Triple-GAN is able to transfer smoothly on the data level with clear semantics.

| (a) Airplane | (b) Bird | (c) Cat | (d) Deer |
| (e) Dog | (f) Frog | (g) Ship | (h) Truck |

Figure 2: Samples from Triple-GAN given certain class on CIFAR10.

| (a) Data | (b) Samples | (c) Linear Interpolation |

Figure 3: (a): randomly sampled MNIST data; (b) disentanglement of class and style; (c) class-conditional interpolation for Triple-GAN on MNIST.

# E   Convergence Speed

Though Triple-GAN has one more network, its convergence speed is at least comparable with Improved-GAN, as presented in Fig. 4. Both the models are trained on SVHN dataset with default settings and Triple-GAN can get good results in tens of epochs. The reason that the learning curve of Triple-GAN is oscillatory may be the larger variance of the gradients due to the presence of discrete variables. Also note that we apply pseudo discriminative loss at epoch 200 and then the test error is reduced significantly in 100 epochs.

# F   Detailed Architectures

We list the detailed architectures of Triple-GAN on MNIST, SVHN and CIFAR10 datasets in Table 1, Table 2 and Table 3, respectively.

Figure 4: Convergence speed of Improved GAN and Triple-GAN on SVHN.

Table 1: **MNIST**

| Classifier C | Discriminator D | Generator G |
|---|---|---|
| Input 28×28 Gray Image | Input 28×28 Gray Image, Ont-hot Class representation | Input Class y, Noise z |
| 5×5 conv. 32 ReLU<br>2×2 max-pooling, 0.5 dropout<br>3×3 conv. 64 ReLU<br>3×3 conv. 64 ReLU<br>2×2 max-pooling, 0.5 dropout<br>3×3 conv. 128 ReLU<br>3×3 conv. 128 ReLU<br>Global pool<br>10-class Softmax | MLP 1000 units, lReLU, gaussian noise, weight norm<br>MLP 500 units, lReLU, gaussian noise, weight norm<br>MLP 250 units, lReLU, gaussian noise, weight norm<br>MLP 250 units, lReLU, gaussian noise, weight norm<br>MLP 250 units, lReLU, gaussian noise, weight norm<br>MLP 1 unit, sigmoid, gaussian noise, weight norm | MLP 500 units,<br>softplus, batch norm<br><br>MLP 500 units,<br>softplus, batch norm<br><br>MLP 784 units, sigmoid |

Table 2: **SVHN**

| Classifier C | Discriminator D | Generator G |
|---|---|---|
| Input: 32×32 Colored Image | Input: 32×32 colored image, class y | Input: Class y, Noise z |
| 0.2 dropout<br>3×3 conv. 128 lReLU, batch norm<br>3×3 conv. 128 lReLU, batch norm<br>3×3 conv. 128 lReLU, batch norm<br>2×2 max-pooling, 0.5 dropout | 0.2 dropout<br>3×3 conv. 32, lReLU, weight norm<br>3×3 conv. 32, lReLU, weight norm, stride 2<br><br>0.2 dropout | MP 8192 units,<br>ReLU, batch norm<br>Reshape 512×4×4<br>5×5 deconv. 256. stride 2,<br>ReLU, batch norm |
| 3×3 conv. 256 lReLU, batch norm<br>3×3 conv. 256 lReLU, batch norm<br>3×3 conv. 256 lReLU, batch norm<br>2×2 max-pooling, 0.5 dropout | 3×3 conv. 64, lReLU, weight norm<br>3×3 conv. 64, lReLU, weight norm, stride 2<br><br>0.2 dropout | 5×5 deconv. 128. stride 2,<br>ReLU, batch norm |
| 3×3 conv. 512 lReLU, batch norm<br>NIN, 256 lReLU, batch norm<br>NIN, 128 lReLU, batch norm<br>Global pool<br>10-class Softmax, batch norm | 3×3 conv. 128, lReLU, weight norm<br>3×3 conv. 128, lReLU, weight norm<br><br>Global pool<br>MLP 1 unit, sigmoid | 5×5 deconv. 3. stride 2,<br>sigmoid, weight norm |

Table 3: **CIFAR10**

| Classifier C | Discriminator D | Generator G |
|---|---|---|
| Input: 32×32 Colored Image | Input: 32×32 colored image, class y | Input: Class y, Noise z |
| Gaussian noise<br>3×3 conv. 128 lReLU, weight norm<br>3×3 conv. 128 lReLU, weight norm<br>3×3 conv. 128 lReLU, weight norm<br>2×2 max-pooling, 0.5 dropout | 0.2 dropout<br>3×3 conv. 32, lReLU, weight norm<br>3×3 conv. 32, lReLU, weight norm, stride 2<br><br>0.2 dropout | MLP 8192 units,<br>ReLU, batch norm<br>Reshape 512×4×4<br>5×5 deconv. 256.stride 2<br>ReLU, batch norm |
| 3×3 conv. 256 lReLU, weight norm<br>3×3 conv. 256 lReLU, weight norm<br>3×3 conv. 256 lReLU, weight norm<br>2×2 max-pooling, 0.5 dropout | 3×3 conv. 64, lReLU, weight norm<br>3×3 conv. 64, lReLU, weight norm, stride 2<br><br>0.2 dropout | 5×5 deconv. 128. stride 2<br>ReLU, batch norm |
| 3×3 conv. 512 lReLU, weight norm<br>NIN, 256 lReLU, weight norm<br>NIN, 128 lReLU, weight norm<br>Global pool<br>10-class Softmax wieh weight norm | 3, ×3 conv. 128 lReLU, weight norm<br>3×3 conv. 128, lReLU, weight norm<br><br>Global pool<br>MLP 1 unit, sigmoid, weight norm | 5×5 deconv. 3. stride 2<br>tanh, weight norm |