[Reviews · NeurIPS 2017]

Reviewer 1



In this paper, the authors propose a new formulation of adversarial networks for image generation, that incorporates three networks instead of the usual generator G and discriminator D. In addition, they include a classifier C, which cooperates with G to learn a compatible joint distribution (X,Y) over images and labels. The authors show how this formulation overcomes pitfalls of previous class-conditional GANs; namely that class-conditional generator and discriminator networks have competing objectives that may prevent them from learning the true distribution and preventing G from accurately generating class-conditional samples. The authors identify the following deficiency in class-conditional GAN setups: “The competition between G and D essentially arises from their two-player formulation, where a single discriminator network has to play two incompatible  roles—identifying fake samples and predicting labels”. The argument goes that if G were perfect, then a class-conditional D has an equal incentive to output 0 since the sample comes from G, and to output 1 since the image matches the label. This might force D to systematically underperform as a classifier, and therefore prevent G from learning to produce accurate class-conditional samples. A question about this: - What if a classifier network and a real/fake network are used instead of combining both decisions into a single net? Does the problem still arise? - In practice, does G ever produce realistic enough samples that the hypothetical situation described above actually arises? In my experience training GANs, the situation is always that D can easily recognize samples as fake, and generally needs to be heavily constrained in order to allow learning to happen at all. Since this argument is very important to motivating the paper, it would be nice to have a clear demonstration of this failure mode on real data, even if it is toy data. The proposed model builds two conditional models - generator P(x | y) and classifier P(y | x). By drawing samples from the marginals P(y) and P(x), they can draw samples from the joint distributions (x,y) parametrized by each of these conditional models, to be fed into the discriminator D.
 The model achieves state-of-the-art semi-supervised learning results most notably on CIFAR-10. Qualitatively, the model achieves clear disentangling of class and style on challenging datasets including SVHN and Cifar-10, and compelling class-conditional samples of automobiles and horses. Overall, I think the proposed model is very interesting and the experimental results are strong, but the paper would benefit from a clearer motivation and illustration of the particular failure modes that it is trying to overcome.

Reviewer 2



Summary: The paper presents a GAN-like architecture called Triple-GAN that, given partially labeled data, is designed to achieve simultaneously the following two goals: (1) Get a good generator that generates realistically-looking samples conditioned on class labels; (2) Get a good classifier, with smallest possible prediction error. The paper shows that other similar GAN-based approaches always implicitly privileged either (1) or (2), and or needed much more labeled data to train the classifier. By separating the discrimination task between true and fake data from the classification task, the paper outperforms the state-of-the-art, both in (1) and (2). In particular, the classifier achieves high accuracy with only very few labeled dataset, while the generator produces state-of-the-art images, even when conditioned on y labels. Quality & Clarity: The paper indeed identifies plausible reasons of failure/inefficiency of the other similar GAN-type methods (such as Improved-GAN). The underlying idea behind Triple-GAN is very elegant and comes with both nice theoretical justifications and impressive experimental results. However the overall clarity of exposition, the text-flow and the syntax can still be much improved. Concerning syntax: too many typos and too many grammatically incorrect sentences. Concerning the exposition: there are too many vague or even unclear sentences/paragraphs (see below); even the statements of some Corollaries are vague (Corollary 3.2.1 & 3.3.1). Some proofs are dismissed as evident or refer to results without precise citation. There are some long and unnecessary repetitions (f.ex. l.58-73 and l.120-129 which are essentially the same), while other points that would need more explanations are only mentioned with a few unclear sentences (l.215-220). This is in striking contrast with the content's quality of this paper. So before publication, there will be some serious work to do on the overall presentation, exposition, and syntax! Detailed comments: l.13-14: the expression 'concentrating to a data distribution', which is used in several places in this paper, does not exist. Please use f.ex. something like 'converges to'. l. 37-38: formulation of points (1) and (2) is too vague and not understandable at the first read. Please reformulate. l. 46-49: good point, but formulation is cumbersome. Reformulate it more neatly. You could f.ex. use a 'on the one hand ... on the other ...' type of construction. l. 52-56: Unclear. ('the marginal distribution of data' , 'the generator cannot leverage the missing labels inferred by the discriminator': what do you mean?). Please reformulate. l.58-73: too many unnecessary overlap with l.120-129. (And text is clearer in l.120-129, I find...). Please mind repetitions. One only needs repetitions if one messed up the explanation in the first place... l. 140&157: please format your equations more nicely. l .171: proof of Lemma 3.1: last sentence needs an explanation or a reference. l.174: Corollary 3.2.1: I find it clearer to say p(x) = p_g(x) = p_d(x) and p(y) = p_g(y) = p_d(y). l.185: Corollary 3.3.1: What is a 'regularization on the distances btw ...' ? Define it. Without this definition, I can't say whether the proof is obvious or not. And in any way, please say one or two sentences why the statement is true, instead of dismissing it as obvious. l.215-220: I don't see what the REINFORCE alrogrithm, which is used in reinforcement learning, has to do with your algorithm. And the whole paragraph is quite obscure to me...Please reformulate. (Maybe it needs to be a bit longer?). Esperiments: - please state clearly when you are comparing to numbers reported in other papers, and when you really re-run the algorithms yourself. l.228: '10 000 samples' should be '10 000 test samples' l.232: '(Results are averaged over 10 runs)' : don't put it in parenthesis: it's important! Also, remind this in the caption of Table 1. l.254-263: how many labeled data do you use for each of the experiments reported here? Please specify! Also, you did not mention that triple-GAN simply also has more parameters than an improved GAN, so it already has an advantage. Please mention it somewhere. table 1: what are the numbers in parenthesis? 1 std? Calculated using the 10 runs? l.271: 'the baseline repeats to generate strange samples' : reformulate. Generally: there are quite a few articles ('the') and other small words missing in the text. A careful proof reading will be necessary before publication. Answer to rebuttal: We thank the reviewers for their answers and hope that they will incorporate all necessary clarifications asked by the reviewers. And please be precise: in your rebuttal, you speak of a distance over joint probability distributions, and then cite the KL-divergence as an example. But mathematically speaking, the KL-divergence is not a distance! So please, be precise in your mathematical statements!

Reviewer 3



This paper proposes a three-player adversarial game to overcome the fact that a discriminator in a semi-supervised setting has two incompatible roles, namely to classify and separate real data from fake data. The paper is well-written and the authors display a good knowledge of the GAN literature. I think it proposes a solution to a relevant problem, and the empirical evidence presented to back up the claims being made is convincing. The major criticism I have to voice is on the way in which the core concepts behind Triple-GAN are presented. The ideas start off clearly, but eventually become less and less principled as heuristic regularizers are added to the loss function. The notion that there exists three players also seems a little arbitrary: the same distinction could be applied to ALI/BiGAN, where the generator encompasses both conditional distributions, but in that case the ALI/BiGAN authors chose not to make the distinction. I feel like the joint distributions would be a much better place to make that distinction. There are three different joint distributions: the “labeled” joint p(x, y), the “unlabeled” joint p(x)p(y | x), and the “generative” joint p(y)p(x | y), and all three need to match. Instead, Triple-GAN folds the “unlabeled” and “generative” joint distributions into a “mixture” distribution with mixture weights alpha and (1 - alpha). In analogy with GAN and ALI/BiGAN, equilibrium is attained when the “labeled” and “mixture” joints match. The equality between the “unlabeled” and “generative” joint is guaranteed if one of the two is equal to the “labeled” joint. This is achieved by explicitly enforcing that the “unlabeled” joint matches the “labeled” joint through maximum likelihood of the “unlabeled” joint on “labeled” joint samples. It feels like these constraints could be alleviated by considering three discriminator classes instead of two, each corresponding to one of the three joint distributions. Additional note: the semi-supervised results presented in Table 1 appear to be inconsistent in the way SOTA is displayed: sometimes Triple-GAN is presented as better than other methods, even though the error bars overlap (e.g., Improved-GAN on MNIST n=100). How are the uncertainties computed? Also, the ALI numbers are out-of-date: for instance, ALI achieves an error rate of 17.99 (±1.62) on CIFAR10, and therefore its error bar overlaps with Triple-GAN.